# A Lightweight Intrusion Detection System with Dynamic Feature Fusion Federated Learning for Vehicular Network Security

**DOI:** 10.3390/s25154622

**Published:** 2025-07-25

**Authors:** Junjun Li, Yanyan Ma, Jiahui Bai, Congming Chen, Tingting Xu, Chi Ding

**Affiliations:** 1School of Electrical and Information Engineering, Zhengzhou University, Zhengzhou 450001, China; mayanyan@gs.zzu.edu.cn (Y.M.);; 2Longmen Laboratory, Luoyang 471000, China; 3State Key Laboratory of Intelligent Agricultural Power Equipment, Luoyang 471000, China

**Keywords:** CAN, intrusion detection, federated learning, autoencoder, CNN, LSTM, attention

## Abstract

The rapid integration of complex sensors and electronic control units (ECUs) in autonomous vehicles significantly increases cybersecurity risks in vehicular networks. Although the Controller Area Network (CAN) is efficient, it lacks inherent security mechanisms and is vulnerable to various network attacks. The traditional Intrusion Detection System (IDS) makes it difficult to effectively deal with the dynamics and complexity of emerging threats. To solve these problems, a lightweight vehicular network intrusion detection framework based on Dynamic Feature Fusion Federated Learning (DFF-FL) is proposed. The proposed framework employs a two-stream architecture, including a transformer-augmented autoencoder for abstract feature extraction and a lightweight CNN-LSTM–Attention model for preserving temporal and local patterns. Compared with the traditional theoretical framework of the federated learning, DFF-FL first dynamically fuses the deep feature representation of each node through the transformer attention module to realize the fine-grained cross-node feature interaction in a heterogeneous data environment, thereby eliminating the performance degradation caused by the difference in feature distribution. Secondly, based on the final loss $L_{AE}\left(X,\hat{X}\right)$ index of each node, an adaptive weight adjustment mechanism is used to make the nodes with excellent performance dominate the global model update, which significantly improves robustness against complex attacks. Experimental evaluation on the CAN-Hacking dataset shows that the proposed intrusion detection system achieves more than 99% F1 score with only 1.11 MB of memory and 81,863 trainable parameters, while maintaining low computational overheads and ensuring data privacy, which is very suitable for edge device deployment.

## 1. Introduction

As consumer demand for advanced functionalities in autonomous vehicles grows, modern cars are increasingly integrating sophisticated sensors and ECUs. This integration directly increases system complexity [1,2]. Modern vehicles commonly use the CAN bus to manage numerous ECUs. The CAN bus employs a broadcast communication mechanism characterized by a low latency, high throughput, and reliable data transmission, effectively reducing the cost and complexity associated with point-to-point ECU connections. However, the standard CAN bus design presents significant security risks. Firstly, as a broadcast network, all nodes can listen to every communication. Secondly, it lacks mechanisms to authenticate the identity of message senders or receivers. Finally, it allows attackers to easily impersonate or forge message sources [1]. With the increasing prevalence of wireless connectivity in vehicles, the security threats facing vehicle networks are becoming more severe. Despite the inherent reliability of CAN buses, effective defensive measures remain inadequate against emerging security threats [3]. Consequently, establishing a robust IDS becomes critical for identifying malicious threats within vehicle networks. While previous IDS research has yielded progress, the increasing complexity of network traffic, evolving attack methods, and limited computing resources continue to pose significant challenges for developing high-performance, lightweight, and easily deployable IDS solutions.

As vehicle networks (IVNs) continue to rapidly evolve, increasingly sophisticated vehicle functionalities require more ECUs, significantly amplifying cybersecurity threats and increasing the complexity and frequency of network intrusion attacks. Traditional intrusion detection methods, such as signature-based approaches, are insufficient for identifying unknown attacks (zero-day attacks) and adapting to evolving attack strategies. Consequently, intrusion detection systems leveraging machine learning (ML) and deep learning (DL) have emerged as vital research areas in network security [4,5]. Traditional ML approaches like K-Means clustering [6], support vector machines (SVM) [7] and extreme gradient boosting (XGBoost) [8,9,10,11,12,13] have been applied effectively by learning patterns from historical data without predefined feature definitions. However, these techniques rely heavily on manual feature engineering, limiting their generalizability and automation [14]. Additionally, conventional ML models struggle with handling complex, high-dimensional data and capturing deeper structural information, ultimately constraining model accuracy. Conversely, DL approaches automatically extract meaningful features directly from raw network data, excelling at identifying deep patterns within complex datasets. Recent studies applying DL methods [15,16,17,18], such as CNN [19,20,21], LSTM [22,23,24,25,26] and AE [27,28,29] have demonstrated promising results. Nevertheless, DL models typically require substantial computational resources, such as GPUs or TPUs, presenting challenges including high training costs, poor real-time performance, and large communication overhead in centralized models. These issues are particularly problematic in edge computing environments and can risk exposing sensitive data [4,30]. Federated learning (FL), a distributed learning approach enabling training across multiple devices without exchanging raw data, effectively addresses these concerns [31,32,33,34,35]. Incorporating federated learning into IDS development thus offers a promising pathway toward lightweight, privacy-preserving intrusion detection frameworks.

This research proposes a novel lightweight IDS model with the ability to accurately detect a variety of known attacks. To reduce computational requirements and ensure user privacy, we introduce a federated learning-based theoretical framework specifically designed for deployment within the DFF-FL environment.

This research offers several innovations compared to existing studies:(1)Previous works mainly integrate autoencoders with other models like recurrent networks, but rarely incorporate attention mechanisms. In contrast, we design an autoencoder module enhanced by multi-head attention and normalization techniques. This design enables the model to capture global dependencies, not just local or temporal patterns. Residual connections and nonlinear mappings further strengthen its ability to extract informative features.(2)The proposed lightweight hybrid model first extracts key features using convolutional layers. These features are then analyzed with recurrent layers to identify temporal patterns. Finally, a self-attention component guides the recurrent layers to emphasize critical regions identified earlier, thereby enhancing predictive performance.(3)Traditional methods typically rely heavily on manual preprocessing, which may overlook deeper hidden patterns in raw data. Our approach adopts a dual-stream structure. One stream retains original data features, while the other employs a Transformer-enhanced autoencoder to discover deeper abstract features from the data.(4)We also introduce a novel federated learning framework that dynamically integrates features across multiple nodes. Unlike traditional federated learning that aggregates only model parameters, our method effectively captures and combines detailed local feature information. By integrating federated learning with autoencoder techniques, we achieve collaborative learning at the feature level while preserving data privacy.

The rest of this paper is arranged as follows. In Section 2, the related work is reviewed and the advantages and disadvantages of the current research are analyzed. In Section 3, the proposed IDS framework is elaborated. Section 4 presents the experimental implementation of the proposed IDS model and the comparative analysis of the results. Section 5 concludes the paper, points out the possible shortcomings of this research and looks forward to future research directions. Section 6 concludes the study and emphasizes its significance.

## 2. Related Work

The rapid development of connected and autonomous vehicles (CAVs) brings increasingly severe security threats. Traditional intrusion detection methods struggle against sophisticated and unknown attacks. This encourages researchers to explore smarter, more adaptive detection methods, particularly those based on machine learning and deep learning.

Alshathri et al. [36] focused on security issues within vehicle networks, particularly the challenge posed by imbalanced data (where normal data significantly outnumbers attack instances). They proposed an intelligent attack detection framework and evaluated different resampling methods. Although some classifiers (e.g., k-NN) achieved perfect metrics after resampling, the authors did not thoroughly discuss limitations or provide detailed validation procedures. Such ideal results raise concerns about potential model overfitting or information leakage during data splitting.

Sun et al. [15] proposed an intrusion detection method by combining a CNN and LSTM. The CNN component extracts spatial and temporal features from images derived from raw data, while the LSTM handles sequential dependencies. This hybrid model successfully distinguishes normal from malicious traffic by automatically capturing both local and temporal patterns, eliminating the need for manual feature extraction. However, the feasibility of deploying this approach in edge computing environments or applying model compression techniques remains unaddressed.

Li et al. [37] developed an intrusion detection system that utilizes a pre-trained CNN model and fine-tunes it under the intrusion detection framework of transfer learning (TL). They employed particle swarm optimization for tuning hyperparameters and combined predictions from multiple CNN classifiers to enhance detection accuracy. On benchmark datasets, their approach achieved detection rates and F1 scores exceeding 99% with minimal prediction latency. However, despite reporting computational costs, the complexities of model integration and parameter optimization could significantly increase deployment expenses. The authors did not thoroughly address strategies for model compression, lightweight implementation, or edge-device deployment.

Anbalagan et al. [38] introduced an intelligent intrusion detection system that converts vehicle network traffic into images, enabling efficient feature extraction through enhanced CNNs. Their approach achieved an accuracy of approximately 98%. Although initial benefits of transforming data into images are demonstrated, the authors did not provide quantitative comparisons with methods relying on raw data features, particularly regarding information retention and computational overhead.

To address resource limitations and privacy concerns in vehicle networks, Mothukuri et al. [4] proposed a multi-stage intrusion detection system based on a hierarchical federated learning framework. Their system performs a rapid initial detection of known attacks and subsequently identifies unknown attacks. Their experimental results showed nearly perfect detection rates, minimal false alarms, and a strong protection of sensitive data through localized processing. This federated approach successfully balances high accuracy and privacy, making it highly suitable for real-world deployment.

A primary research challenge is creating an efficient intrusion detection architecture suitable for resource-limited devices that can detect known and unknown attacks while ensuring data privacy. Although deep learning models offer powerful performance, their high computational requirements remain problematic. Therefore, optimizing model structures and integrating distributed training methods (such as federated learning) have emerged as critical research objectives. Motivated by these issues, this study develops an intrusion detection strategy specifically targeting common attack scenarios on CANs. Our goal is to enhance vehicle communication security, effectively defend against sophisticated attacks, and ensure reliable operation of connected and autonomous vehicles.

The main contributions of our current research work are as follows:

A dual-stream architecture is developed, enabling the model to simultaneously extract abstract features using an autoencoder (branch A) while preserving original data features (branch B).

An improved autoencoder branch integrating the Transformer mechanism is designed to automatically extract the hidden features in the vehicular network, avoiding the prior bias caused by manual feature engineering and the redesign of the feature set, so as to improve the feature expression ability.

This work is the first to apply a lightweight CNN-LSTM–Attention variant to vehicular-network intrusion detection. By operating directly on raw traffic with one-dimensional convolutional neural network (1D CNN), the model preserves temporal information and avoids the costly image conversion required by two-dimensional convolutional neural network (2D CNN). This design markedly improves attack detection accuracy while remaining compact enough for deployment on edge devices.

We introduce a DFF-FL framework. Unlike conventional federated learning, which aggregates only model parameters, DFF-FL also merges feature-space representations from individual nodes. A Transformer-based attention module assigns adaptive weights to these features, enabling the global model to capture fine-grained differences among heterogeneous data sources.

## 3. Methods

This section details the related methods proposed in our research work. The overall flow chart of the model is shown in Figure 1. Firstly, the improved autoencoder based on a Transformer captures the long-term dependence between data for the preprocessed data containing time series information, so as to enhance the performance of the model on time series data. Then, the lightweight variant model is used to strengthen the feature expression in the abstract features while capturing the local pattern in the original input data. The attention mechanism is introduced to make the LSTM component pay more attention to the important part of the CNN reconstruction features in the final prediction, so as to improve the prediction performance of the model. Finally, the theoretical framework of DFF-FL for model deployment is expounded.

### 3.1. Data Processing

The CAN Hacking Dataset is a real-world traffic dataset focusing on in-vehicle CAN bus network attacks. It is characterized by capturing a variety of typical attacks against the CAN protocol mixed with normal traffic data. However, there are some problems such as high noise, imbalanced class distribution, and a non-standardized data format (hexadecimal payloads and variable-length frames). Therefore, a preprocessing step must be performed to eliminate noise interference, construct temporal correlation features, and mitigate sample bias, so as to provide high-quality inputs for the subsequent training of intrusion detection models.

#### 3.1.1. Data Normalization

Due to varying feature scales, raw data can cause numerical instability and slow convergence during model training. To solve this problem, we first reconstruct all window data into a two-dimensional matrix, then linearly map each feature to the interval [0, 1], and finally re-transform it into a three-dimensional tensor. The mapping formula (1) is as follows:(1)$$X_{norm}\;=\;\frac{x\;-\;\;x_{min}}{x_{max}\;-{\;x}_{min}}\;,$$
where $x_{min}$ and $x_{max}$ are the minimum and maximum values of the feature column. Normalizing the inputs to a common range stabilizes optimization and enables the network to exploit its parameters more efficiently.

#### 3.1.2. Data Augmentation

To enhance the robustness of the algorithm to noise and data shifts, the training data are augmented with random noise. Gaussian noise with a mean of zero and adjustable standard deviation σ is added to the input features to realize batch and efficient sample expansion, so as to enhance the generalization ability of the autoencoder during training, as shown in Equation (2):(2)$$X\;={\;X}_{norm}\;+\;\varepsilon ,\;\varepsilon \sim N\left(0,\sigma^{2}\right),$$
selecting *σ* as a hyper-parameter allows the model to tolerate noise without unduly distorting the underlying feature structure.

#### 3.1.3. Sliding Time Windows

To effectively capture temporal characteristics, the original data is segmented into overlapping time windows. Each window has a length of five time steps, with a step size of eight. Each window segment is labeled based on whether it contains attack instances or only normal data. For each window, the dimension of the feature matrix is (T, F). If any frame in the window has Flag = 1, it is considered an attack; otherwise, it is considered normal. This can be formally described using Equations (3) and (4):(3)$$X_{i}=\left\{X_{i},X_{i+1},\ldots ,X_{i+W-1}\right\},\;\forall i\in \left\{1,1\;+\;s,1\;+\;2s,\ldots \right\},$$(4)$$y_{i}=\left\{\begin{array}{c} 0,\;\;\;\;\;\;\;\;\;\;\;\;if\;\;Flag=0\\ a_{\dot{J}}+1,\;\;if\;Flag=1 \end{array}\;\;\right.,$$
where $X_{i}$ denotes the i windowed segment and $y_{i}$ its corresponding label.

### 3.2. Improved Autoencoders with Transformers

The traditional autoencoder model mainly consists of an encoder and a decoder. The encoder compresses the input data layer by layer through a series of fully connected or convolutional layers to extract local features, and the decoder reconstructs the low-dimensional representation back to the original space. When dealing with data with long-range dependence such as time series data, such models often rely only on fixed weights and local convolutions, which makes it difficult to capture the global correlation across time steps in the series, and their simple structure easily leads to overfitting and an insufficient generalization ability. To this end, this paper introduces the multi-head attention mechanism of Transformer into the encoder. This improvement has the following four main advantages:(1)*Modeling Long-Range Dependencies:* Transformer models excel at capturing long-range dependencies in the input data, and are particularly well suited for dealing with time series data. Traditional autoencoders progressively compress and decode data through fully connected layers or convolutional layers, which may not effectively capture long-range dependencies between data. After introducing the multi-head attention mechanism, the Transformer can focus on different parts of the input data at the same time, thereby enhancing the feature extraction ability.(2)*Adaptive Weight:* Through the multi-head attention mechanism, the Transformer model assigns different weights to each input feature during the encoding process, which enables the model to dynamically adjust the focus according to the context information and improves the expression ability of the feature. However, the traditional autoencoder only relies on the encoder layer with fixed weights, which limits its ability to learn complex data structures.(3)*Enhanced Feature Representation:* The feature representation ability of the model is significantly improved by introducing Transformers. In time series tasks, Transformer can learn the internal structure of input data from multiple perspectives through the parallel computing of multiple attention heads, and this multi-dimensional learning way can often help the model to better identify different types of patterns or anomalies.(4)*Improved Robustness:* Since the Transformer model handles complex data through residual connections and attention mechanisms, the improved autoencoder remains more stable during training and generalizes better when facing various types of data changes. This is particularly important in anomaly detection, where anomalous data often deviate from regular patterns, requiring the model to be adaptable and robust.

Autoencoders aim to minimize reconstruction errors without relying heavily on labeled data, making them highly adaptable to scenarios with limited data availability. Encoding raw data into a lower-dimensional form reduces noise and emphasizes essential features relevant to normal or abnormal behaviors. Traditional autoencoders mainly rely on local information for feature extraction, but fail to consider the long-range dependence between data. When dealing with data containing time series information, long-term dependencies are often critical to the performance of the model. Unlike traditional autoencoders, we introduce Transformer-based attention mechanisms into the encoding process. This attention enables parallel modeling of relationships across multiple data segments, capturing both local patterns and broader, long-term dependencies. Such global modeling capabilities allow the autoencoder to identify anomalies distributed throughout entire sequences, not just isolated events. By incorporating global context into low-dimensional representations, the Transformer-enhanced module significantly improves downstream anomaly detection accuracy.

The core structure of the current module is shown in Figure 2, and consists of an input layer followed by a multi-head attention block that model dependencies across all time steps in parallel. The formula is given in (5):(5)$$Attn\left(Q,K,V\right)=softmax\left(\frac{Q\cdot K^{T}}{\sqrt{d_{k}}}\right)\cdot V,$$
where $Q=K=V=X\in R^{T\times F}$.

Features are compressed to dimension size α with the help of a fully connected layer with ReLU activation. A decoder layer with sigmoid activation then reconstructs the data to its original shape. The autoencoder is trained to minimize reconstruction error. To balance overall fidelity and anomaly sensitivity, the final loss $L_{AE}\left(X,\hat{X}\right)$ is determined by the weighted sum of the mean squared error (MSE) and the mean absolute error (MAE). This joint loss preserves global accuracy while improving sensitivity to outliers. The specific process is shown in Equations (6)–(11) below:(6)$$z\;=\;ReLU\left(P\cdot W_{e}\;+\;b_{e}\right),\;\;W_{e}\in R^{F\;\times \;d},$$(7)$$\hat{X\;}=\sigma \cdot \left(z\cdot W_{d}+b_{d}\right),\;\;\;\;\;W_{d}\in R^{d\times F},$$(8)$$E=Encoder\left(\hat{X}\right)\;\in R^{T\times k},$$(9)$$L_{AE}\left(X,\hat{X}\right)\;=\;MSE\left(X,\hat{X}\right)\;+\;MAE\left(X,\hat{X}\right),$$(10)$$MSE\;=\;\frac{1}{N}\sum_{i\;=\;1}^{N}{\left\|X_{i}\;-{\;\hat{X}}_{i}\right\|}_{2}^{2},$$(11)$$MAE\;=\;\frac{1}{N}\sum_{i\;=\;1}^{N}{\left\|X_{i\;}-\;{\hat{X}}_{i}\right\|}_{1},$$
where $X\in R^{T\times F}$ represents the input, and $W_{e},W_{d}$ represent the parameters of encoding and decoding, respectively. $P\in R^{T\times F}$ is the intermediate representation obtained by adding the attention projection result and the original input through the residual, and then normalizing the layer. It not only retains the information from the original input, but also fuses the global context extracted by self-attention. $z\in R^{F\times k}$ is the encoder output, which is the low-dimensional latent code at each time step. It is concentrated the most critical features in P, which are used for subsequent decoding and classification. $\hat{X}$ is the decoder output, and $L_{AE}\;\left(X,\hat{X}\right)\;$ is denoted as the loss function (final error).

The above improved model implementation Algorithm 1 is shown in the following table.


**Algorithm 1.** Transformer-Enhanced Autoencoder ConstructionInput: XOutput: Branch A: Encoder (E); $\hat{\;X},\;L_{AE}$1: procedure Transformer-enhanced Autoencoder2: Input Layer ← Input3: The intermediate feature A is calculated according to the multi-head attention mechanism4: Layernormalization Layer yields intermediate variables *p*5: Compute Encoder output z according to Equation (6)6: Compute the reconstruction matrix $\hat{X}$ according to Equation (7)7: Calculate the Latent feature matrix E according to Equation (8)8: Compute the reconstruction loss $L_{AE}$ according to Equations (9)–(11)9: end procedure


### 3.3. Lightweight CNN-LSTM-Attention Variant Model

A pioneering study first applied a CNN-LSTM–Attention model to intrusion detection and reported strong results. Most follow-up work, however, still converts traffic data into images and processes them with 2D CNN [39]. This approach has two key drawbacks. (1) Artificial spatial bias: Mapping raw sequences to grayscale images imposes an assumed spatial structure, obscuring true temporal patterns and adding redundant computation. (2) Heavy parameter load: A standard 2D convolution layer contains kernel height × kernel width × input channels × filters parameters, quickly inflating model size.

Unlike previous approaches, our method utilizes a double-layer 1D convolutional operations on both streams. This naturally preserves the local sequential structure between adjacent data points without manual feature engineering. The convolutional layers reduce feature dimensions while effectively capturing local patterns.

To effectively combine abstract features extracted by the autoencoder with the original multi-scale input features, we propose a dual-stream architecture. The detailed structure is shown in Figure 3. These two streams are integrated using an adaptive feature fusion mechanism. Branch A employs a pretrained autoencoder to reduce input dimensionality and extract critical abstract features. These features are further refined through convolution and pooling operations, resulting in a concise feature representation. Branch B directly processes the original sequential input using similar convolutional operations to capture contextual information. The specific process is shown in Formulas (12)–(15) below:(12)$$C_{A}\;=\;MaxPool\;(ReLU(W_{a2}\;*\;ReLU\left(W_{a1}\;*\;E\;+\;b_{a1}\right)\;+\;b_{a2})),$$(13)$$P_{A}=ReLU\;\left(W_{pa}\;\cdot \;C_{A}+{\;b}_{pa}\right),$$(14)$$C_{B}=MaxPool\;(ReLU\;(W_{b2}\;*\;ReLU\left(W_{b1}\;*\;X+b_{b1}\right)+b_{b2})),$$(15)$$P_{B}=ReLU\;(W_{pb}\;\cdot \;C_{B}+b_{pb}),$$
where $\hat{X}$ represents the decoder output, $W_{a1},W_{a2},b_{a1},b_{a2},\;W_{b1},W_{b2},b_{b1},b_{b2}$ are all 0 convolutional neural network parameters, and MaxPool (ReLU (·)) represents the convolution and pooling process, $C_{A,}C_{B}$ represents the timing feature after convolutional pooling. $P_{A},P_{B}$ denote the final feature matrix mapped to a 32-dimensional space.

The adaptive fusion mechanism integrates outputs from both branches using weights learned automatically by a trainable neural network. Specifically, anomaly scores derived from global statistical features in branch A dynamically determine how the two feature sets are combined. This adaptive approach effectively highlights anomalies and preserves essential context, significantly enhancing detection performance. The formula for the adaptive fusion process is given in (16)–(18) below:(16)$$\bar{P_{A}}\;=\;\;\frac{1}{T^{\prime }d^{\prime }}\sum_{i,j}{\left(P_{A}\right)}_{i,j}\;\;\;\;,$$(17)$$\alpha =\sigma \;\cdot \;\left(W_{\alpha }\;\cdot \bar{\;P_{A}}+b_{\alpha }\right),\;\;\;\in \left(0,1\right),$$(18)$$S=\alpha \;\cdot {\;P}_{A}+\left(1-\alpha \right)\;\cdot \;P_{B},$$
where $\bar{P_{A}}$ is the average value of the feature from branch A, representing the aggregated feature over the entire time window, $T^{\prime }$ is the length of the time window, and $d^{\prime }$ is the dimension number of the feature. α is dynamically adjusted according to the average feature of the input, which highlights the sensitivity of the model to abnormal features, and S represents the final fused feature.

The fused features of branch A (abnormal features) and branch B (original features) are further modeled by stacking the two-layer LSTM to extract high-order temporal dependencies. To mitigate overfitting, dropout regularization is applied, and attention mechanisms aggregate temporal features, guiding the decision-making process. This comprehensive, multi-scale feature learning strategy significantly improves the model’s robustness and interpretability, providing an innovative solution for detecting complex anomalies in practical scenarios. The Bahdanau layer is applied to the two-layer LSTM output $H\in R^{T\cdot u}$ to make the classifier focus on the most informative time step. The context vector c is fed into the fully connected layer and then the final attack classification is performed with the help of the Softmax layer. The flow is given in Equations (19)–(25):(19)$$H^{\left(1\right)}\;=\;LSTM\left(S\right),$$(20)$$H^{\left(2\right)}=LSTM\left(H^{\left(1\right)}\right),$$(21)$$e_{t}=V^{T}\;\cdot \;tan\;h\left(W_{1}\;\cdot \;H_{T^{\prime }}^{\left(2\right)}+{\;W}_{2}\;\cdot {\;H}_{t}^{\left(2\right)}\right),$$(22)$$\alpha_{t}=\frac{\exp\left(e_{t}\right)}{\sum_{u}\exp\left(e_{u}\right)},$$(23)$$c=\sum_{i}\alpha_{t}\;\cdot {\;H}_{t}^{\left(2\right)},$$(24)$$y=softmax\left(W_{0}\;\cdot \;c+b_{0}\right),\;\;\;\;y\in \;R^{C},$$(25)$$L_{c}=-\sum_{k=1}^{K}y_{k}\;\cdot \;\ln\hat{y_{k}},$$
where $H_{T^{\prime }}^{\left(2\right)}$ is the hidden state at time t.$\;H_{t}^{\left(2\right)}$ is the final hidden state, and $V,W_{1},W_{2}$ are learnable parameters, α denotes the attention weight, c is the context vector, and $L_{c}$ is the cross-entropy loss, c denotes the number of classification labels, and $y_{k},\hat{y_{k}}$ represent the one-hot ground-truth label and the predicted probability for class k, respectively.

### 3.4. Dynamic Feature Fusion Federated Learning (DFF-FL)

The rapid growth of data storage and computation significantly boosts the development of artificial intelligence across various industries. However, data governance still faces major challenges related to privacy protection and regulatory compliance. Additionally, isolated and sensitive datasets limit effective sharing and collaboration. Federated learning, a promising technology, addresses these challenges by enabling collaborative modeling while ensuring data privacy [40].

FL is a decentralized machine learning approach designed to overcome the challenges associated with isolated and sensitive data. By enabling multiple local nodes to collaborate with one or more central servers, federated learning supports distributed modeling without moving data away from its source. Traditional federated learning primarily aggregates model parameters and often overlooks deeper feature-level information present across local nodes. A comparison with previous federated learning techniques is shown in Table 1. To address this gap, this study introduces a Dynamic Feature Fusion Federated Learning (DFF-FL) framework, outlined as follows in Figure 4. Blue block denotes the local feature extraction in local nodes, where the strategies in Section 3.1, Section 3.2 and Section 3.3 are adopted. Green block represents the dynamic feature fusion in edge servers. Yellow block represents the iterative convergence in the central server, where the data transmitted from the edge servers is updated until convergence, and the updated global features and global model parameters are returned to the local nodes. 

#### 3.4.1. The DFF-FL Step

(1)***Local Feature Extraction.*** Each local node independently performs the following steps. First, data preprocessing includes segmenting data into time windows, enhancing and normalizing it. Next, a Transformer-based autoencoder is trained locally using normal samples, generating abstract features. Finally, each node trains a classification model with labeled data (normal and attack samples). Nodes with better detection performance are dynamically assigned greater weights.(2)***Dynamic Feature Aggregation.*** A central server gathers encoded features from each node. Features are combined adaptively, based on reconstruction errors from anomaly detection results. Nodes with a higher detection accuracy are assigned larger aggregation weights. This adaptive aggregation process uses attention mechanisms to highlight contributions from high-performing nodes. The fused global features are then broadcast back to local nodes for further training. In real-world vehicular networks or Internet of Things (IoT) environments, factors such as varying data volumes, imbalanced data distributions (normal vs. attack samples), network quality, and available computing resources may lead to significant fluctuations in local model performance after each training round. We aim to ensure that nodes performing well on known attacks have a larger influence on the global model, while nodes with poorer performance contribute less, thereby preventing suboptimal updates from degrading overall model performance. As federated training progresses, the evaluation metrics of each node evolve.

In the DFF-FL framework, suppose there are N active nodes at the *i*-th node and the quality score of the *i*-th node is $S_{i}^{\left(t\right)}$ (obtained based on $L_{{AE}_{i}}$), as shown in Equation (26).(26)$$S_{i}^{\left(t\right)}=\frac{1}{1+L_{{AE}_{i}}},$$

To maintain accurate “voting weights”, we define the parameter $\beta_{t}$ to control the smoothness of the weight distribution. Firstly, the server normalized each $\beta_{t}$ based on the standard deviation of the node quality score $S_{i}^{\left(t\right)}$ to obtain the initial weight, as shown in Equations (27) and (28)(27)$$\beta_{t}=k\;\cdot {\;\sigma }_{S_{t}}^{-1},$$(28)$$\;\alpha_{i}^{\prime }=\frac{S_{i}}{\sum_{j=i}^{N}S_{i}},$$

A larger node performance gap makes the higher performance node weight more concentrated. The adaptive weights are recalculated after each round to reflect the current local performance. Quality scores are converted to regularization weights $\alpha_{i}^{\left(t\right)}$ by Equation (29). The global aggregated features and global model parameters are then calculated based on the dynamic weights, as shown in Equations (29) and (31)(29)$$\alpha_{i}^{\left(t\right)}=\frac{e^{{\beta \cdot S}_{i}^{\left(t\right)}}}{\sum_{j=1}^{N}e^{{\beta \cdot S}_{j}^{\left(t\right)}}},$$(30)$$f_{global}^{\left(t\right)}=\sum_{i\;=\;1}^{N}\alpha_{i}^{\left(t\right)}\;\cdot {\;E}_{i}^{\left(t\right)},$$(31)$$\theta_{global}^{\left(t\right)}=\sum_{i\;=\;1}^{N}\alpha_{i}^{\left(t\right)}\;\cdot {\;\theta }_{i}^{\left(t\right)},$$
where $\beta_{t}$ controls the smoothness of the weight distribution, $\sigma_{S_{t}}$ is the standard deviation of the node quality score of the current federated round, k is a hyperparameter, typically in the range [0.5, 1]. $S_{i}$ is the anomaly detection performance score (final loss function $\;L_{AE}$) of node i; normalization is performed on $\alpha_{i}^{\prime }$. Let $\alpha_{i}$ denote the attention weight of node i; $E_{i}^{\left(t\right)}$ denotes the feature-level aggregation, the encoded feature of node i; let $\theta_{i}^{\left(t\right)}$ denote the aggregation of model parameters, that is, the model parameters of node i; $f_{global}^{\left(t\right)}$ represents the global aggregated feature. $\theta_{global}^{\left(t\right)}$ denotes the global aggregation parameter.

(3)***Local Model Updating.*** The aggregated global features $f_{global}^{\left(t\right)}$ and model parameters $\theta_{global}^{\left(t\right)}$ are redistributed back to each node, and the features are added to the local training for a new round of local model training, which further promotes the unification of cross-node feature space, strengthens the transfer of global knowledge to the local model, and improves the overall federated learning performance. In particular, each node updates its local model and reevaluates the reconstruction error after receiving the global features from the central server. The error of these updates determines the aggregation weights in subsequent training rounds.

(4)***Iterative Convergence.*** The described three-step process iterates continuously until the federated model reaches convergence. Model convergence is evaluated based on detection accuracy for known attacks and generalization ability for unknown threats.

#### 3.4.2. Convergence Analysis

In order to ensure the stable convergence of federated learning, DFF-FL adopts the following convergence criteria, as shown in Equation (32):(32)$$\frac{{\left\|\theta_{global}^{\left(t\right)}-\theta_{global}^{\left(t-1\right)}\right\|}_{2}}{{\left\|\theta_{global}^{\left(t-1\right)}\right\|}_{2}}\;\;<\;\;\varepsilon_{\theta }$$The F1-score is used as the key performance index, and the model convergence is determined when the performance improvement of K consecutive rounds is less than the threshold $\varepsilon_{\theta }$ (usually 10^–3^).

#### 3.4.3. Analysis of Data Heterogeneity and Communication Overhead

In typical federated learning application scenarios such as vehicular networks or the Internet of Things, each participating node often has different data scales, category distribution and collection conditions. At the same time, the communication between nodes depends on the wireless network, and the delay and bandwidth fluctuate significantly. How to efficiently and robustly train the global model under the premise of ensuring data privacy is a key issue to be solved urgently. Current data heterogeneity manifests in two forms:(1)***Statistical Heterogeneity****:* The local data distribution of each node is not independent and identically distributed (non-IID), for example, some nodes only contain normal traffic, while others may only contain specific types of attack samples.(2)***System Heterogeneity***: This mainly refers to the uneven computing power, storage and energy consumption capabilities of nodes. To address the above data heterogeneity problem, we propose a theoretical framework of federated learning through Dynamic Feature Fusion (DFF-FL), namely an adaptive weight adjustment mechanism. The autoencoder is used to efficiently extract the corresponding latent features without damaging the data privacy of the nodes, and the contribution of each node is dynamically weighted based on the reconstruction error in the global model iteration process, which ensures that the high-performance nodes contribute more weight and suppresses the negative impact of low-quality updates, so as to alleviate the training instability caused by heterogeneous data, improving federated learning performance.

In the DFF-FL mechanism, each federated learning round involves the following communication: model parameters (θᵢ): 81,863 parameters with a storage size of hundreds of KB; feature representation (Eᵢ): low-dimensional sparse features generated by the Transformer–autoencoder, KB in size.

It is assumed that in the actual vehicle Wi-Fi (20 Mbps) environment, the round-trip transmission delay of the above data is controlled within 50 ms, which obviously meets the requirements of real-time application scenarios of the internet of vehicles (V2X communication delay < 100 ms).

## 4. Results

### 4.1. Evaluation

In this work, the Car–Hacking dataset is selected [41]. It consists of four files, each corresponding to a distinct attack type: denial of service, fuzzy attack, spoofing drive gear, and spoofing RPM gauge. Each file contains both injected and normal messages. During preprocessing, any rows with missing values are removed. The dataset includes the following features: timestamp, CAN ID, DLC, DATA [0–7], and Flag. Previous research has mentioned that there is a strong correlation between timestamps and mock attack intervals; Therefore, we excluded timestamp features at the beginning of our analysis. Hexadecimal values (such as CAN ID and DATA fields) are converted into numerical values for further analysis. Additionally, the ‘Flag’ field, initially labeled as transmitted (T) or received (R), is transformed into binary values: transmitted messages labeled as 1 and received as 0. This binary encoding simplifies subsequent feature engineering for attack detection. The specific formula is shown in (33):(33)$$Flag\;=\;\left\{\begin{array}{c} 0,\;\;\;\;\;\;\;\;\;if\;\;Flag\;=\;R\\ 1,\;\;\;\;\;\;\;\;if\;Flag\;=\;T \end{array}\right.,$$

A central tool for evaluating the performance of classification models is the Confusion Matrix, where the row labels represent the True class, the column labels represent the Predicted Label, and the cell values represent the number of examples in which the true class is the row label and the predicted class is the column label.

To evaluate the robustness of the results of the proposed model, we used k-fold cross validation. Metrics derived from the confusion matrix are used to characterize model performance, specifically True Positives (TP), False Positives (FP), True Negatives (TN), and False Negatives (FN). The autoencoder is re-initialized at the start of each fold to ensure independent feature learning. The trained encoder is combined with the main classifier to classify the validation set, and the four metrics of accuracy, precision, recall and F1-score are used to measure the model performance. The details are given in Formulas (34)–(37). The mean and standard deviation of these metrics across the ten folds are calculated to measure stability and generalization capabilities.(34)$$\;Accuracy=\frac{TP+TN}{TP+TN+FP+FN}\;,$$(35)$$Precision=\frac{TP}{TP+FP},$$(36)$$Recall=\frac{TP}{TP+FN},$$(37)$$F1=2\times \frac{Pre\times Rec}{Pre+Rec},$$

### 4.2. Analysis of Model Parameter Setting

In this experiment, we adopt an orthogonal experimental design, fix other hyperparameters, adjust only the single factor by one, and record the F1 score and average delay in each fold through five-fold cross validation to determine the optimal parameter configuration. Specifically, for the time window length T, when *T* takes the value of 3, 5, 8, or 12, T=5 can make the F1 score reach the peak, and further increasing it does not bring significant improvement. Considering that T<5 may lead to insufficient information, and T>10 will cause the inference delay to increase by an O(T^2^) level, aggravate the noise filling, and dilute the effective signal, T=5 is finally selected. On this basis, grid search is used to optimize other hyperparameters for the condition T=5. The results show that when the noise standard deviation σ=0.02, Transformer–AE can learn smoother feature representations. However, when σ>0.05, the signal-to-noise ratio decreases and the model performance begins to degrade. In the multi-head attention mechanism, the number of heads h=2 CAN capture two CAN frame modes of ID space and data bytes in parallel. Increasing h>2  increases the number of parameters by about 95%, but does not significantly improve the F1 score. The key dimension key_dim=8 can ensure the stability of Query·Key scaling. If it continues to increase, it easily leads to overfitting. Finally, we verify the influence of the parameters obtained from the above grid search on the model performance through simulation experiments, and the results are summarized in Table 2. The overall parameter settings of the final model are shown in Table 3.

The optimal dimensionality of the autoencoder’s latent vector E is determined with the Elbow Method [42]: the reconstruction error is plotted against candidate dimensions (1–32), and the “elbow” point—where the marginal gain in error reduction tapers off—is identified at nine dimensions (See Figure 5). This choice balances representational capacity and computational overhead and is applied throughout the experiments.

### 4.3. Model Performance Results and Analysis

We conducted a model evaluation of the proposed intrusion detection system on the CAN–Hacking dataset and compared it with recent studies using the same data (As shown in Table 4). First of all, a comparison of the model was carried out, that is, the model with a single branch structure only retaining the data flow of branch A was used for the experiment. Although the input branch of the original data was proposed, satisfactory results were still obtained. The overall metrics of the single-stream structure are slightly lower than those of the two-stream structure, but both the single-stream structure and the two-stream architecture achieve an F1 value of more than 99%, which exceeds the competitive methods in the attack classification task.

To assess robustness and generalization, we conducted 10-fold cross-validation. Figure 6 reports per-fold metrics and presents confusion matrices for the dual-stream model and a single-stream baseline. The dual-stream system misclassifies two DoSs and 106 fuzzy-attack messages as normal, while it achieves nearly perfect accuracy for gear- and RPM-spoofing attacks. The single-stream model produced a similar error pattern, confirming the overall stability of both approaches. The residual false negatives for DoS and fuzzy traffic likely arise from feature overlap with normal messages and the absence of class-balancing during training. Despite this, the dual-stream IDS demonstrates a strong overall detection capability and near-perfect recognition of gear-spoofing attacks and RPM-spoofing attacks, underscoring its effectiveness in multi-class intrusion detection.

### 4.4. Model Complexity

Considering the limited memory, computing power, and bandwidth constraints of on-board ECUs [48], a lightweight architecture is essential for the deployment of IDS. Therefore, we characterize the model complexity by measuring the model size and the number of trainable parameters. The model comparison is shown in Table 5. To ensure both efficiency and high performance, we simplified the proposed model structure, reducing its size significantly. Despite employing a dual-stream design, careful optimization of layer depth and neuron count resulted in a compact model size of 1.11 MB. Additionally, the number of trainable parameters directly affects the speed of training and inference processes, and models with fewer parameters generally perform these tasks faster [49]. Our Transformer-based autoencoder combined with the lightweight CNN-LSTM–Attention model consists of 81,863 trainable parameters. The total training time is approximately 2257.678 s, with inference per data packet taking only 0.036 s. The experimental results demonstrate that our proposed deep-learning-based IDS effectively detects network attacks while being sufficiently lightweight for practical deployment on edge devices.

As mentioned above, the overall parameter number of the model is in the tens of thousands, and the model size of the two-branch model is about 1.11 MB, which fully meets the requirements of Flash storage (usually ≥ 2 MB) and RAM (starting from a few MB) of the vast majority of on-board ECUs. A single sample (time window T = 5, feature dimension d = 10, LSTM unit h = 64) is used for theoretical FLOPs estimation (as shown in Table 6):

In theory, a single frame inference of the whole model requires about <0.1 MFLOPs. Even on low-clock-rate MCUS (e.g., 200 MHz Cortex-M4), inference can be performed in milliseconds with floating-point or fixed-point hardware acceleration.

The current number of Transformer headers is two, which can capture both ID space and data byte dependencies. If the number Transformer headers continue to increase (taking headers = 4 as an example), the Query/Key/Value will expand across the board, which will eventually be reflected in an approximately 95% increase in the number of parameters. The results of the current experimental theory stage have confirmed that h = 2 and key_dim = 8 are close to the optimal cost performance. In practice, however, increasing the number of heads to 4 or 12 to be more expressive is a resource trade-off.

Finally, regarding the analysis of inference time, it should be noted that this experiment measures the average inference time of a single sample using the pure Python 3.8 CPU inference model, which may be constrained by the current experimental conditions. In fact, the final deployment environment can be further optimized by several times to tens of times, fully meeting the millisecond level detection requirements.

## 5. Discussion

In this paper, we propose a lightweight intrusion detection system (IDS) for vehicular network security, which significantly improves the anomaly detection performance of vehicular networks through an innovative dynamic Feature fusion Federated Learning (DFF-FL) framework. Specifically, the innovation of this paper is mainly reflected in the following aspects:

Firstly, a two-stream architecture is proposed to extract abstract features using a Transformer-enhanced autoencoder, while capturing the temporal and local patterns of the data through a lightweight CNN-LSTM–Attention model. This two-stream design not only effectively avoids the prior bias that may be brought by traditional manual feature engineering, but also improves the sensitivity and detection accuracy to complex attacks.

Secondly, this paper innovatively applies the dynamic feature fusion mechanism to federated learning. Traditional federated learning methods only aggregate at the model parameter level, failing to make full use of the rich information at the feature level between nodes. The DFF-FL framework proposed in this paper adaptively fuses the feature expressions of different nodes through the Transformer attention mechanism, which significantly enhances the global model’s ability to capture the fine-grained differences in heterogeneous data sources.

Although the method proposed in this paper shows significant advantages in theory, there are still some shortcomings in the research. Firstly, the current research mainly focuses on the known attack types, and lacks the detection ability experiments for unknown attacks (zero-day attacks), which may limit the protection effectiveness of the system against emerging attacks in practical applications. In addition, this paper has not carried out simulation experiments, and only stays in the theoretical framework stage and the experimental verification stage using a small amount of real data, lacking in-depth analysis and evaluation of the operating performance in the actual network environment.

Therefore, future research can be further deepened in the following aspects. On the one hand, the detection ability of unknown attack types could be increased, and the generalization ability of the model to unknown threats could be improved by introducing unsupervised learning or semi-supervised learning mechanisms. On the other hand, more comprehensive simulation experiments and actual deployment tests should be carried out to evaluate the real-time performance, stability and robustness of the model in the real environment. In addition, a more refined adaptive weight adjustment mechanism can be explored to further optimize the efficiency and effectiveness of feature fusion between nodes in the federated learning framework. These future research directions will contribute to the further application and promotion of the IDS method proposed in this paper in actual vehicular network security protection.

## 6. Conclusions

Our study proposes a robust, efficient, and privacy-preserving IDS for vehicular network security by innovating the integration of a transformer enhanced autoencoder, a lightweight CNN-LSTM-Attention architecture, and DFF-FL. The dual-stream architecture effectively captures both abstract and original feature information, addressing limitations found in traditional methods reliant on manual feature engineering. The adaptive feature fusion approach significantly enhances model sensitivity to anomalies, improving detection accuracy. Federated learning integration ensures privacy protection and a computational efficiency suitable for resource-constrained environments. Experimental results confirm the method’s superior performance, achieving a near-perfect detection accuracy with minimal inference latency and computational demands. Future research directions include further optimization of the adaptive weighting mechanisms and extending the framework to detect unknown or evolving threat patterns in broader autonomous vehicle network scenarios.

## Figures and Tables

**Figure 1 sensors-25-04622-f001:**
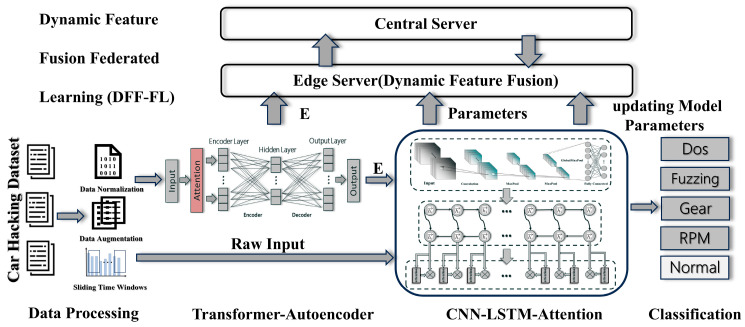
Two-branch IDS architecture.

**Figure 2 sensors-25-04622-f002:**
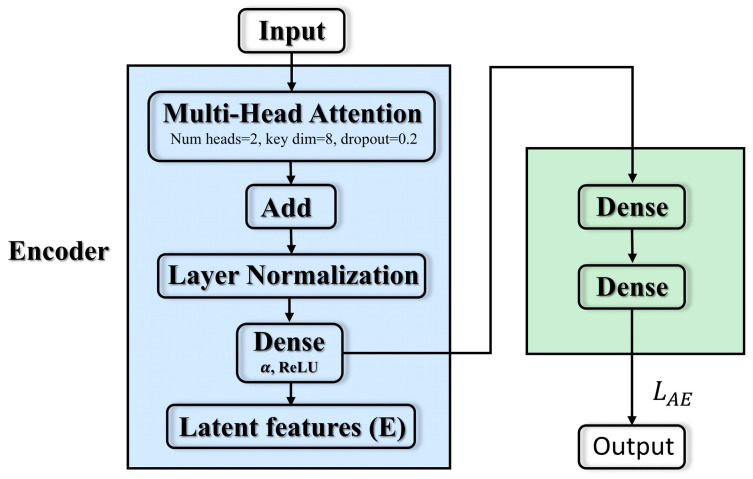
Transformer–autoencoder.

**Figure 3 sensors-25-04622-f003:**
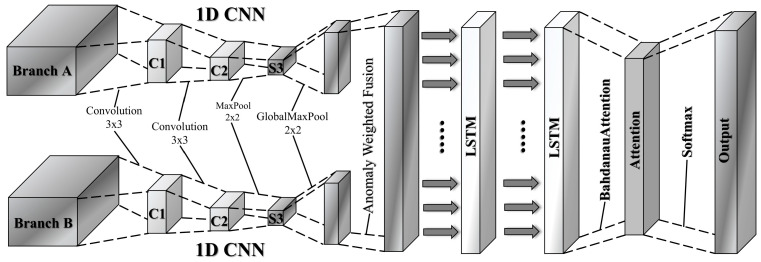
Lightweight CNN-LSTM–Attention variant model neural network architecture.

**Figure 4 sensors-25-04622-f004:**
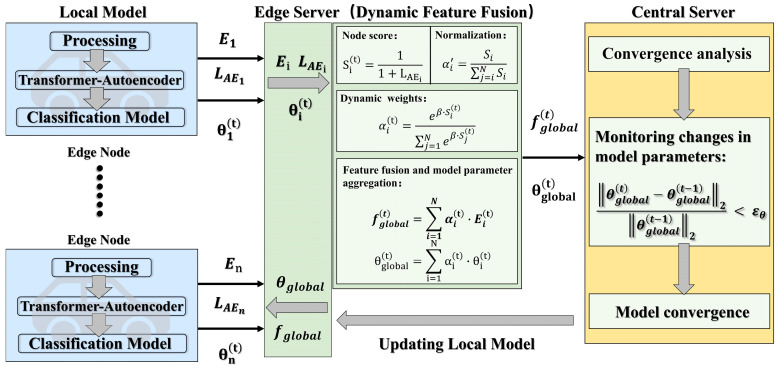
Theoretical framework of Dynamic Feature Fusion Federated Learning (DFF-FL).

**Figure 5 sensors-25-04622-f005:**
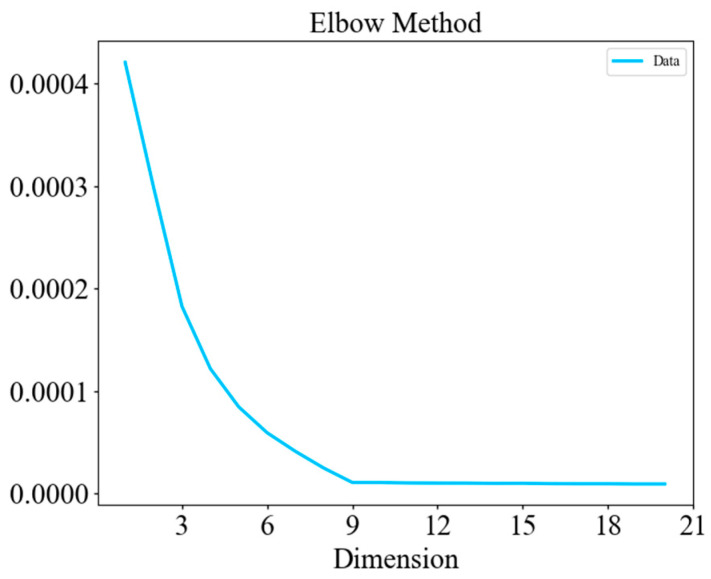
Determination of the AE latent feature dimension.

**Figure 6 sensors-25-04622-f006:**
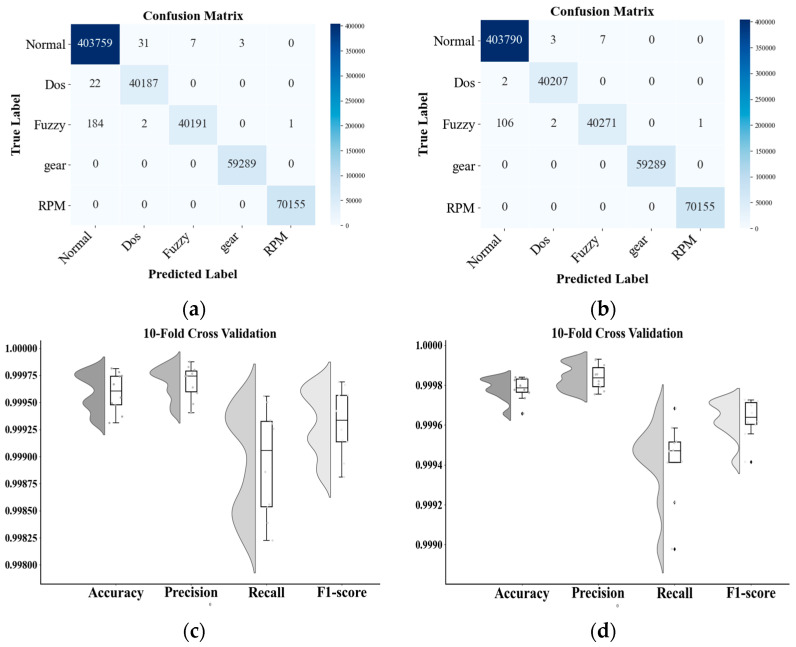
Confusion matrix and 10-fold cross-validation results: (**a**) single-branch architecture model CM; (**b**) single-branch architecture model CM; (**c**) single-branch cross-validation results; (**d**) two-branch cross-validation results.

**Table 1 sensors-25-04622-t001:** Comparison of theoretical frameworks.

Core Link	Traditional FL(H-FL [30])	DFF-FL
Feature extraction method	Purely Local Model Training	Local Feature Extraction for Transformer-Autoencoder
Aggregation mechanism	Parametric average Aggregation	Attention-based Dynamic Feature Aggregation
Weight adjustment	Fixed or static weights	Adaptive dynamic weights based on anomaly detection performance
Applicable scenarios	Homogeneous data environment	Highly heterogeneous data environments

**Table 2 sensors-25-04622-t002:** Parametric analysis.

K_FOLDS = 5	F1-Score	Average Latency
T = 3	0.9997 ± 0.0000	36.44 ms
**T = 5**	**0.9998 ± 0.0001**	**35.25 ms**
T = 8	0.9998 ± 0.0001	35.6 8ms
T = 12	0.9998 ± 0.0000	35.62 ms
σ = 0.0	0.9995 ± 0.0001	21.17 ms
σ = 0.01	0.9997 ± 0.0001	21.37 ms
σ ** = 0.02**	**0.9998 ± 0.0001**	**21.04 ms**
σ = 0.05	0.9997 ± 0.0001	21.38 ms
σ = 0.1	0.9997 ± 0.0001	21.43 ms
h = 1	0.9996 ± 0.0003	21.41 ms
**h = 2**	**0.9997 ± 0.0001**	**21.37 ms**
h = 4	0.9997 ± 0.0001	21.52 ms
key_dim = 4	0.9996 ± 0.0001	21.22 ms
key_dim ** = 8**	**0.9997 ± 0.0003**	**21.24 ms**
key_dim = 16	0.9997 ± 0.0001	21.38 ms

**Table 3 sensors-25-04622-t003:** Detailed model parameter settings.

Modules	Parameters	Setting Values
Processing	T	5
	Sliding step size	8
Data Augmentation	σ	0.02
	Chunk size	1024
Transformer-Autoencoder	Head of attention	2
	key dimension	8
	Encoding dimension	9
	Dropout rate	0.2
CNN	Number of convolution kernels	32
	Kernel size	3
	Pooling size	2
LSTM	Number of LSTM cells	64
	Number of LSTM layers	2
	LSTM Dropout	0.2
Training parameters	Batch Size	64
	Initial learning rate	0.001
	Early Stopping Patience	5

**Table 4 sensors-25-04622-t004:** Performance comparison of models on CAR-HACKING DATASET.

Method	Accuracy (%)	Precision (%)	Recall (%)	F1 (%)
P-LeNe [43]	98.10	98.14	98.04	97.83
ID-CNN [44]	99.96	99.94	99.63	99.80
LSTM [45]	-	99.9	99.9	99.9
DCNN [46]	99.93	99.84	99.84	99.91
ACGAN [47]	-	99.23	99.24	99.23
Our (single branch)	99.96	99.97	99.90	99.93
**Our (dual branch)**	**99.98**	**99.98**	**99.94**	**99.96**

**Table 5 sensors-25-04622-t005:** Model comparison.

Method	Model Parameters	Model Size (MB)	Training Time (s)	Test Time Per Packet (s)
ANN+LSTM-AE [30]	253,582	2.98	-	-
AE-GAN [49]	2.15 Million	-	-	-
MTH-IDS [2]	-	2.61	-	-
**Our (single branch)**	**76,709**	**1.01**	**2091.931**	**0.035**
Our (dual branch)	81,863	1.11	2257.678	0.036

**Table 6 sensors-25-04622-t006:** Theoretical FLOPs estimation.

Modules	Complexity	Numerical Magnitude
Self-Attention	O(T^2^·d)	$5^{2}\;\times$ 10 = 250 scalar operations
Projection & LayerNorm	O(T·d^2^)	5 × 10^2^ = 500
Dense (latent→recon)	O(T·d·z)	5 ×10 × 9 = 450
1D Conv (2 layers × 32 channels)	O(T·k·Cin·Cout)	5 ×3 ×9 ×32 × 2 ≈ 8640
LSTM (2 layer, h = 64)	O(T·(h + d)·h) × 2	5 ×(64 + 32) ×64 × 2 ≈ 61,440
Attention Merging & Dense Head	O(h^2^) + O(h·C)	$64^{2}\;+\;64\;\times$ 5 ≈ 4256 + 320
Total	≈> 75,000 FLOPs	

## Data Availability

The dataset analyzed during the current study is publicly available. The dataset can be accessed via the following link: [https://ocslab.hksecurity.net/Datasets/CAN-intrusion-dataset] (accessed on 22 July 2025).

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
