# Peer review of "A Lightweight Intrusion Detection System with Dynamic Feature Fusion Federated Learning for Vehicular Network Security"

_sensors, 2025, doi:10.3390/s25154622_

Round 1
Reviewer 1 Report
Comments and Suggestions for Authors
This paper proposes a novel lightweight Intrusion Detection System (IDS) for vehicular networks using Dynamic Feature Fusion Federated Learning (DFF-FL). The IDS employs a dual-stream architecture that combines a Transformer-based autoencoder for deep feature extraction with a CNN-LSTM-Attention model to capture temporal and local patterns. By dynamically aggregating features from distributed nodes using attention-based weighting, the system enhances anomaly detection accuracy.
Suggested Improvements to the Article:
-
A Related Work section must be included to contextualize the contributions within existing literature.
-
Please expand Subsections 2.1 and 2.2 with more detailed explanations and technical clarity.
-
Equations 7, 8, 9, 10, and 11 require elaboration to improve understanding of the underlying mathematical formulations.
-
Algorithm 1 (Line 174) and Table 1 are unclear. Please provide further explanation or clarification for better reader comprehension.
-
Figure 4 should be enhanced to improve its clarity and usefulness. Additionally, the text in all figures is too small and should be increased for readability.
-
Expand Section 3.2 (Model Complexity, Line 345) with more comprehensive analysis, including computational efficiency, scalability, and inference time.
Reviewer 2 Report
Comments and Suggestions for Authors
General Comments:
This manuscript introduces a novel intrusion detection system (IDS) tailored for vehicular networks, integrating a Transformer-enhanced autoencoder with a lightweight CNN-LSTM-Attention model, coupled with a unique federated learning approach named Dynamic Feature Fusion Federated Learning (DFF-FL). The system is evaluated using the CAN-Hacking dataset, achieving notable accuracy with a low computational footprint suitable for resource-constrained devices.
Overall, the work addresses a relevant and timely topic, considering the increasing cybersecurity threats associated with vehicular networks and edge computing. However, certain aspects require further clarification and improvement.
Authors are required to address the following (minor issues):
- The abstract can further clarify the novelty and significance of the federated learning approach (DFF-FL) explicitly, emphasizing why it is advantageous over traditional methods.
- Similarly, the introduction can be enhanced by clearly stating the specific contributions in a structured manner (e.g., bullet points), highlighting exactly how this research advances beyond previous studies (Lines 78 – 95).
- Although the proposed model is generally well-described, additional details regarding hyperparameters (e.g., transformer attention heads, exact CNN/LSTM architecture parameters, training epochs, optimizer) would significantly enhance reproducibility.
- Similarly, the DFF-FL mechanism requires further clarity in terms of communication overhead and convergence analysis. Clearer explanations or diagrams of the adaptive weight adjustment mechanism would strengthen technical comprehension.
- Authors are required to provide deeper insights into the sensitivity analysis of their hyperparameters (e.g., window size, noise augmentation levels, transformer parameters).
- Moreover, considering federated learning scenarios, a discussion of data heterogeneity and network delays in real-world scenarios would add significant practical value.
- The manuscript lacks practical deployment analysis, such as experiments on real edge devices (e.g., Raspberry Pi or NVIDIA Jetson Nano). Including these details would substantiate claims regarding real-time applicability and lightweight performance.
- Figures and tables could benefit from enhanced clarity with improved legends and consistent formatting.
- Proofreading for consistency in tense, subject-verb agreement, and punctuation would enhance readability.
Comments on the Quality of English Language
- Proofreading for consistency in tense, subject-verb agreement, and punctuation would enhance readability.
